# Cover Crops Modulate the Response of Arbuscular Mycorrhizal Fungi to Water Supply: A Field Study in Corn

**DOI:** 10.3390/plants12051015

**Published:** 2023-02-23

**Authors:** Micaela Tosi, Cameron M. Ogilvie, Federico N. Spagnoletti, Sarah Fournier, Ralph C. Martin, Kari E. Dunfield

**Affiliations:** 1School of Environmental Sciences, University of Guelph, 50 Stone Rd. E, Guelph, ON N1G 2W1, Canada; 2Department of Plant Agriculture, University of Guelph, 50 Stone Rd. E, Guelph, ON N1G 2W1, Canada; 3Instituto de Investigaciones en Biociencias Agrícolas y Ambientales (INBA), Consejo Nacional de Investigaciones Científicas (CONICET), Avda. San Martín 4453, Buenos Aires C1417DSE, Argentina; 4Cátedra de Microbiología, Facultad de Agronomía, Universidad de Buenos Aires, Avda. San Martín 4453, Buenos Aires C1417DSE, Argentina

**Keywords:** Glomeromycota, 18S rRNA, Illumina sequencing, service crops, drought

## Abstract

Cover crops (CCs) were found to improve soil health by increasing plant diversity and ground cover. They may also improve water supply for cash crops by reducing evaporation and increasing soil water storage capacity. However, their influence on plant-associated microbial communities, including symbiotic arbuscular mycorrhizal fungi (AMF), is less well understood. In a corn field trial, we studied the response of AMF to a four-species winter CC, relative to a no-CC control, as well as to two contrasting water supply levels (i.e., drought and irrigated). We measured AMF colonization of corn roots and used Illumina MiSeq sequencing to study the composition and diversity of soil AMF communities at two depths (i.e., 0–10 and 10–20 cm). In this trial, AMF colonization was high (61–97%), and soil AMF communities were represented by 249 amplicon sequence variants (ASVs) belonging to 5 genera and 33 virtual taxa. *Glomus*, followed by *Claroideoglomus* and *Diversispora* (class Glomeromycetes), were the dominant genera. Our results showed interacting effects between CC treatments and water supply levels for most of the measured variables. The percentage of AMF colonization, arbuscules, and vesicles tended to be lower in irrigated than drought sites, with significant differences detected only under no-CC. Similarly, soil AMF phylogenetic composition was affected by water supply only in the no-CC treatment. Changes in the abundance of individual virtual taxa also showed strong interacting effects between CCs, irrigation, and sometimes soil depth, although CC effects were clearer than irrigation effects. An exception to these interactions was soil AMF evenness, which was higher in CC than no-CC, and higher under drought than irrigation. Soil AMF richness was not affected by the applied treatments. Our results suggest that CCs can affect the structure of soil AMF communities and modulate their response to water availability levels, although soil heterogeneity could influence the final outcome.

## 1. Introduction

Water supply is essential for agroecosystems, as it regulates biological processes belowground (e.g., carbon and nutrient cycling) and aboveground (e.g., plant productivity). While rainfall varies seasonally and interannually within regions, in many of them climate change is leading to heavier but less frequent rainfall events, with extended dry periods [1,2]. Agricultural practices such as cover crops (CCs) and diverse crop rotations are able, to some extent, to help cope with extreme or unsuitable environmental conditions for crop growth [3]. Cover crops, especially those with multiple species, can improve water availability for cash crops by reducing evaporative water loss and increasing soil water storage and capacity, hence making crops more resilient during seasonal droughts [4,5]. Additionally, CCs can reduce erosion risks, enhance carbon sequestration, and reduce nutrient losses and greenhouse gas emissions [6,7,8].

Cover crops may also affect subsequent cash crop growth, and its response to environmental factors, via changes in soil- and plant-associated microbial communities [9,10,11]. This includes arbuscular mycorrhizal fungi (AMF), widespread symbionts of terrestrial plants with widely known beneficial effects on nutrient acquisition and water use efficiency [12]. Because agricultural practices can shift AMF composition and diversity, they may also affect the benefits provided to plants through AMF symbiosis [13,14,15]. In this sense, increasing plant cover and diversity with CCs could not only shift the soil AMF composition but also increase AMF diversity [13]. In fact, previous studies have found shifts in root, rhizosphere, and soil AMF communities of cash crops in response to different CC species [16,17,18]. Yet, only certain CC species, in particular mixtures, were found to increase AMF richness [19]. In addition, the AMF colonization of cash crops was found to be higher in the presence of certain CC species [17,20,21,22]. A meta-analysis by Bowles et al. [13] concluded that such increases in AMF colonization are more likely when using legume CC species and reducing the tillage intensity. Increases in soil AMF propagules and biomass were also found after incorporating certain CC species [20,21,23,24,25,26]. However, such positive effects of CCs on AMF biomass or propagules were sometimes transient and did not persist until the end of the subsequent cash crop [27,28].

Arbuscular mycorrhizae are able to alleviate drought stress in plants [2] by increasing the volume of soil explored, thereby improving water and nutrient acquisition. They may also increase water retention via improved soil aggregation [1,29]. At the same time, AMF communities and AMF colonization levels can be affected by environmental conditions, including water availability. Community shifts can be expected since different AMF species can be more or less adapted to different water supply or availability levels [30,31]. Several studies have reported shifts in AMF community composition, either in soil or root-associated communities, in response to changes in water regimes of different durations [30,32,33,34,35]. Yet, this is not the case in all published studies [36,37], and it is less clear how water supply can affect AMF diversity. Low precipitation levels or drought had negative [37,38], neutral [33,34,36,39], and even positive [35] effects on soil- and root-associated AMF diversity. Similarly, increased precipitation in semiarid environments decreased or did not affect soil AMF diversity, as measured using pyrosequencing [34,40]. Reported effects of water supply or availability on AMF spore germination and root colonization are also inconsistent. While drought reduced AMF colonization in several studies [21,35,41,42,43], neutral or positive effects have also been reported [38,44], sometimes depending on the AMF species, plant species, or nutrient availability [31,37].

The impact of CCs and water availability on soil- and root-associated AMF communities, as well as AMF colonization of cash crops, is not always consistent and seems to be the result of several interacting factors. At the same time, interactions are expected between these two factors. Still, the amount of field studies inquiring about the response of AMF to agricultural practices and climate change is limited [2], especially those using high-throughput sequencing. Therefore, our goal was to study how cover cropping and water supply affected the AMF colonization of corn roots and soil AMF diversity and composition. Our study was carried out over one growing season of a corn (*Zea mays* L.) field trial with and without CCs grown the previous fall. The CC treatment was a mixture of four mycorrhizal plant species: rye (*Secale cereale*), crimson clover (*Trifolium incarnatum*), Japanese millet (*Echinochloa esculenta*), and sunflower (*Helianthus annuus*). During the corn growing season, two water supply treatments were applied within each CC treatment: simulated drought and irrigation (i.e., ambient rainfall with supplemental irrigation). Both AMF colonization and diversity were measured at the end of the trial, when corn plants had reached maturity. The diversity and composition of soil AMF communities was analyzed using high-throughput sequencing (HTS) of the 18S rRNA gene (small subunit or SSU), while AMF colonization of corn roots was measured using microscopy. These measurements were related to corn biomass and yield measurements, as well as the soil mineral N content.

We hypothesized that both water supply and CCs could induce shifts in the composition of native soil AMF communities, with the latter also promoting AMF diversity by introducing more plant species into the rotation. Prolonged drought could also reduce colonization levels relative to the irrigated treatments, as long as waterlogging did not occur in the latter. The potential effects of water supply on both soil AMF communities and AMF colonization could be buffered in soils previously planted with CCs. This could occur, for example, via CC soils harboring a larger and more diverse AMF community, which may be more resistant or resilient to changes in environmental conditions. Additionally, CCs could increase soil moisture retention in drought treatments, decreasing the gap between water supply treatments. Finally, we expected AMF colonization to be positively associated with corn growth and yield, considering the beneficial effects of these symbiotic fungi on plant growth.

## 2. Results

### 2.1. Corn AMF Colonization and Yield

Microscopy analyses showed that the AMF colonization of corn roots, as well as the percentage of arbuscules and vesicles, responded similarly to the water supply (Figure 1). Even though the analysis of variance (ANOVA) detected water supply effects on all three variables (Appendix A), multiple comparisons revealed that these were only clear under no-CC. Compared to drought, irrigation reduced the percentage of AMF colonization (78.5% vs. 90.3%, *p* = 0.008), arbuscules, (53.5% vs. 71.4%, *p* = 0.013), and vesicles (8.8% vs. 20.8%, *p* = 0.005) in no-CC sites (Figure 1A–C). In CC, only AMF colonization was marginally lower under irrigation than drought (85.6% vs. 90.5%, respectively, *p* = 0.062). Cover crops did not affect AMF colonization, arbuscules, or vesicles of the corn roots (Figure 1, Appendix A). Under irrigation, AMF colonization tended to be higher in CC than no-CC (84.6% vs. 78.5%, respectively), but these differences were not significant. Still, under irrigated conditions, roots in CC sites tended to have higher average colonization, arbuscles, and vesicles than those in no-CC (Figure 1).

Corn yield (Mg ha^−1^) was, on average, 45% higher under irrigation than under drought, regardless of the CC treatment (*p* < 0.001) (Appendix A). Under drought conditions, corn yield was 63.7% higher in no-CC than CC (*p* = 0.024) (Appendix A). Similarly, yields per plant were higher in irrigation than drought sites, both with and without CC (*p* < 0.001 and *p* = 0.011, respectively), although CC effects under drought were only marginally detected (*p* = 0.0547) (Appendix A). Corn stover biomass was also higher under irrigation than drought in both CC and no-CC (*p* < 0.001 and *p* = 0.029, respectively), but it was not affected by the CC treatments (Appendix A). Both the corn yield (Pearson r = −0.79, *p* < 0.001) and stover biomass (Pearson r = −0.87, *p* < 0.001) were negatively correlated with AMF root colonization.

### 2.2. Soil AMF Diversity and Composition

After denoising, dereplicating, filtering chimeras, and removing unclassified reads, we recovered a total of 794,733 AMF reads belonging to 249 amplicon sequence variants (ASVs) (496–506 bp long after trimming primers). The percentage of AMF to nontarget reads ranged between 57.6–99.5% (median = 89.7%) and was higher at 10–20 than 0–10 cm (*p* = 0.024) (Figure 2A). We also found that at 0–10 cm, the percentage of the AMF reads tended to be lower in irrigated than drought-affected soils, and lower in no-CC than CC soils (Figure 2A). Alpha rarefaction curves (ASV richness) from all samples reached a steady plateau at 3000 reads, suggesting AMF diversity was completely surveyed at the rarefaction depth chosen for the alpha and beta diversity analyses (10,000 reads). 

Soil AMF evenness (ASV level) was higher in CC than no-CC soils (*p* = 0.027) and higher under drought than irrigation (*p* = 0.029) (Figure 2B), with no interaction between these factors or with soil depth. Compared to drought, irrigation also reduced the variability between sites for the same treatment (Figure 1B). No clear changes were detected in phylogenetic or ASV richness (Appendix A). Alpha diversity metrics calculated at the virtual taxon (VT) or phylogroup level were not sensitive to any of the treatments (data not shown).

When we analyzed the phylogenetic ASV composition of soil AMF communities, we found a contrasting response when using a quantitative or a qualitative metric (weighted or unweighted UniFrac, respectively). Weighted UniFrac detected a small but significant effect of CCs on community composition (R^2^ = 5.61%, F = 3.81, *p* = 0.028) (Appendix A). Using this metric, water supply effects were only clear in the no-CC soils, with 15.9% of the variability explained (F = 8.60, *p* = 0.002) (Figure 3A, Appendix A). In soils under CC, the effect of water supply was not clear (R^2^ = 6.28%, F = 2.56, *p* = 0.085) (Figure 3B, Appendix A). Unweighted UniFrac, on the other hand, was only sensitive to the CC treatments (R^2^ = 6.98%, F = 5.03, *p* = 0.001) (Appendix A). Finally, it is worth mentioning that, using both distance metrics, field block effects on the ASV composition were strong (Appendix A) and, in some cases, seemed to modulate CC and water supply effects (Figure 3 and Appendix A). Analyses at the VT level, using Bray–Curtis and Jaccard distances, produced similar but less clear results as those found at the ASV level (data not shown).

### 2.3. Soil AMF Taxa

In these soils, we detected a total of 32 AMF VTs belonging to the classes Glomeromycetes, Paraglomeromycetes, and Archaeosporomycetes. The majority of these VTs were from the order Glomerales (class Glomeromycetes), with 17 *Glomus* spp. (Glomeraceae), 7 *Claroideoglomus* spp. (Claroideoglomeraceae), and 4 *Diversispora* spp. (Diversisporaceae) (Figure 4A). We also detected two VTs from the genus *Archaeospora* (family Archaeosporaceae, order Archaeosporales, and class Archaeosporomycetes) and two from *Paraglomus* (family Paraglomeraceae, order Paraglomerales, and class Paraglomeromycetes) (Figure 4A). Notably, 22 of the total 32 VTs detected (68.8%) were present in all treatments (Figure 4B). Some VTs were uniquely present in a single treatment, such as *Diversispora* VTX00061 in no-CC under drought and other four VTs in no-CC under irrigation (*Archaeospora* sp., *Glomus* VTX00114, *Glomus* VTX00212, and *Glomus* VTX00219) (Figure 4B). Unique VTs for CC under drought were *Diversispora* Varela-Cervero15 BD1, *Claroideoglomus* VTX00056, and *Claroideoglomus* VTX00340, while CC under irrigation did not present any unique VTs (Figure 4B). We also found two VTs that were excluded from one of the CC treatments: *Glomus* VTX00135 from CC under drought and *Archaeospora* VTX00005 from CC under irrigation (Figure 4B).

In terms of AMF genera prevalence, *Glomus* and *Claroideoglomus* were present in all samples (Appendix A). Contrarily, *Archaeospora* had the lowest abundance and prevalence, being completely absent in CC under irrigation and only present in two out of eight samples in the rest of the CC × water treatments (Appendix A). *Paraglomus* was more prevalent in the no-CC plots, especially under drought, where it was present in all samples (Appendix A). Finally, the prevalence of *Diversispora* was highest in no-CC under drought and lowest in no-CC under irrigation (Appendix A). A summary PCA and correlation analyses of the clr-transformed genera abundances evidenced that *Glomus* and *Claroideoglomus* were strongly correlated, and both genera were negatively correlated with *Paraglomus* (Appendix A). Those soils with higher *Glomus* and *Claroideoglomus*, as well as lower *Paraglomus*, also presented lower phylogenetic diversity, which is probably explained by the higher phylogenetic distance between *Paraglomus* (class Paraglomeromycetes) and the majority of the other genera (class Glomeromycetes) (Appendix A).

Taxa bar plots showed that the frequency (%) of *Paraglomus* (mostly *Paraglomus* sp. but also VTX00239) increased with irrigation, particularly under no-CC (Figure 4C and Appendix A). Some degree of spatial heterogeneity was evidenced as variability between field replicates (Appendix A), in particular a replicate that also behaved differently in terms of the ASV composition (Appendix A). However, the differential abundance testing did not detect any changes at the genus or class level (data not shown). At the VT level, we found strong interactions between CC, water supply, and depth in both taxa frequency (Figure 4C) and abundance changes (Figure 5). It is worth noting that most of the sensitive taxa were of average or higher relative abundance (clr), and the seven lowest abundant taxa (top of heatmap) were not affected by any of the treatments (Figure 5). Six of these lower abundance taxa were uniquely present in one CC × water treatment, with the exception of *Archaeospora* VTX00005, which was present in all treatments except CC under irrigation (Figure 4B). The other two unique VTs (*Claroideoglomus* VTX00056 and *Glomus* VTX00114) are not present in the heatmap because they were filtered out before the analysis due to their low prevalence (i.e., each only present in one site).

The extent to which CCs affected VTs almost always depended on the water supply levels and/or soil depth (Figure 5). The exception was *Claroideoglomus* Yoshimura 13b Glo17, which was consistently higher in CC than no-CC. Comprising both drought and irrigated soils, each CC treatment promoted eight VTs. The CC treatment favored two *Claroideoglomus*, two *Diversispora,* and four *Glomus* VTs, while no-CC favored *Claroideoglomus* VTX00402, *Paraglomus* sp., and six *Glomus* VTs (Figure 5). Among these, *Diversispora* sp., *Diversispora* VTX00060, and *Glomus* VTX00199 showed the strongest response, but they only increased with CC in irrigated soils (Figure 5). Contrarily, *Glomus* VTX00222 was more strongly increased by CC under drought, with an opposite and weaker effect under irrigation. The response of individual VTs to water supply depended on the CC treatment in all cases. Overall, irrigation favored seven VTs (*Paraglomus* VTX00239, two *Diversispora*, and four *Glomus* VTs) and drought nine VTs (*Diversispora* sp. and seven *Glomus* VTs) (Figure 5). The most sensitive taxa to water supply were *Diversispora* sp. and *Diversispora* VTX00060, increasing with irrigation under CC, and *Glomus* VTX00199, increasing with drought under no-CC (Figure 5). Two VTs (*Diversispora* sp. and *Glomus* VTX00151) exhibited a clearly different pattern of response to water supply depending on the CC treatment (Figure 5). Similarly, in CC, *Glomus* VTX00130 responded to irrigation in opposite ways depending on the soil sampling depth (Figure 5).

## 3. Discussion

Overall, the studied soils presented AMF communities with relatively low richness levels (249 ASVs, 32 VTs, and 5 genera). A lower AMF diversity could be expected in agricultural soils, since they are subjected to higher physical disturbance, higher agrochemical inputs, and lower overall plant coverage and diversity than natural ecosystems [45,46]. Soil AMF diversity may have also been underestimated, to some extent, because amplicon-based sequencing only targeted a fraction of the 18S or SSU rRNA gene. A study using a different technology to sequence a larger fragment, extending from the 18S rRNA to the 28S rRNA gene, found a total of 955 ASVs from 12 genera in agricultural soils [36]. The colonization of corn roots by AMF was also relatively high (61–97%), which could be expected for this crop at reproductive growth stages [47,48]. We did not quantify the abundance of AMF in soils, but we used the percentage of AMF reads as a reference value, since the number of AMF 18S rRNA reads has been positively associated with fungal biomass (measured with the AMF-specific fatty acid C16:1*cis*11) [49]. Here, the percentage of AMF reads was not clearly affected by the evaluated treatments, although some trends were observed at 0–10 cm (i.e., positive effect of drought and CCs). This contradicts previous reports of CCs increasing said AMF lipid biomarker [27], although reports also show strong plant species-specific and temporal variability [23,26].

In this study, the inclusion of multispecies CCs in the rotation did not affect AMF colonization of corn roots (Figure 1), contrary to the vast majority of the published research [17,20,21,22,48]. Mycorrhizal CCs are expected to increase AMF colonization of cash crops compared to fallow treatments, mostly via increases in AMF propagules in soil, as AMF are obligate symbionts and need to associate to living plants to obtain C and reproduce [13,23]. Even though a similar trend was observed in irrigated soils, this was not statistically significant. Cover crop mixtures based on AMF host plants, such as the one tested here, may be more beneficial for AMF colonization than single-species CCs [23]. Yet, the lack of leguminous species in our CC mix could partly explain its limited effects in this trial [13]. A more plausible explanation for the lack of response in our study could be that all these plots were under a diverse crop rotation and manure application, in which case the relative effect of CCs on soil AMF communities could be expected to be smaller. Differences between CC treatments may also become less clear later in the growth cycle, especially for crops that tend to have a high mycorrhizal affinity [19]. Finally, the short duration of the trial could also be relevant, although positive effects of CCs on AMF colonization of cash crops have also been observed in short-term trials [22,48].

Despite their negligible influence on AMF colonization, CCs modulated the effect of water supply on this variable (Figure 1). Irrigation reduced AMF colonization of corn roots, including the percentage of arbuscules and vesicles, but this response was only clearly detected under no-CC. This negative response of AMF colonization to higher water availability contradicts several studies where colonization was higher in well-watered systems compared to drought [21,35,37,41,42,43]. However, the response of AMF colonization to water supply or availability is not consistent between studies and could depend on factors such as the duration and actual moisture content of the analyzed treatments [2,44,45]. The reason why differences were only detected under no-CC could be explained by a higher abundance of AMF propagules following CCs, which has been observed in several studies [20,21,23,24,25,26]. This hypothesis is supported by the slightly higher percentage of AMF reads at 0–10 cm (Figure 2).

Soil AMF communities were more sensitive to CCs than AMF colonization. Incorporating CCs had a small but significant impact on soil AMF phylogenetic composition, both in terms of presence-absence and relative abundance of ASVs (Appendix A). These results are similar to previous reports of shifts in AMF community composition in response to CCs [16,17,18]. Because all plant species in the CC mixture were mycorrhizal, they likely associated with the soil AMF and affected their growth and reproduction, hence shaping the soil AMF community. In terms of alpha diversity, CCs did not affect soil AMF richness, but they did increase their evenness, contrary to findings by Njeru et al. [19]. Likely, planting four different plant species in the previous season promoted different AMF more evenly by providing different potential hosts. The conditions in no-CC seem to have favored only a reduced number of ASVs, especially the most dominant one, classified as *Paraglomus* sp. The lack of CC effects on AMF richness has been more frequently reported for soil and root AMF communities [16,25,50], but different CC species could exert different controls over this variable [24]. In our study, the observed small changes in the soil AMF composition and diversity could be explained, as discussed earlier, by the short duration of the trial and/or the fact that it was a relatively diverse crop rotation with a history of manure application [9]. It is also possible that CC practices do not shape soil AMF communities as strongly as other management practices (e.g., tillage, rotation, and cash crop) or environmental factors [51].

As observed for root colonization, water supply effects on soil AMF community composition were also modulated by the CC treatments (Figure 3). We initially hypothesized that this could occur if CCs promoted more abundant and diverse soil AMF communities, which was observed to some extent (Figure 2). It is also possible that CC soils had a smaller difference in soil moisture between drought and irrigated systems. With some exceptions [36], previous studies found that water supply or availability can cause shifts in both soil and root AMF community composition [30,32,33,34]. Such shifts were expected considering AMF species can have different adaptations to the water supply [30,31]. However, as observed here, the response of AMF to water supply seems to be complex, depending on other variables such as previous crops (including CCs), nutrient availability, plant genotype, and growth stage [32,35,37]. In this dataset, interacting effects between CCs and soil properties were clear when comparing community shifts between field blocks (Figure 3). Differences between AMF communities under drought and irrigation may have also been stronger in earlier crop stages for both CC treatments [37], or they could have become stronger over several years. Still, notably, AMF ASV evenness was consistently higher under drought, independent of the CC treatment. This was, at least partly, a consequence of the three most abundant ASVs (classified as *Paraglomus* sp., *Glomus* VTX00063, and *Glomus* sp.) being more dominant in irrigated than drought soils (Figure 4C). It is worth noticing that lower evenness was associated with lower AMF colonization (Appendix A), which could suggest that more dominant taxa had a lower affinity with the cash crop.

*Glomus* and *Claroideoglomus* (family Glomeraceae) were the dominant and most prevalent genera in these soils. These two genera have been found to dominate in many different ecosystems, including agricultural and maize-cropped soils [16,51,52,53]. This is likely due to certain morphophysiological traits found among Glomeraceae (e.g., high growth, high sporulation rate, high turnover rate, and hyphal anastomosis), which enhance their proliferation in many ecosystems, including those under higher disturbance rates such as agricultural soils [54,55]. Furthermore, previous studies found *Glomus* to dominate in slightly acidic soils such as the ones in this study [56]. The lower relative abundance of *Paraglomus* and *Diversispora*, relative to other soil AMF taxa, also matches previous reports on agricultural soils [16].

As observed for other variables, the response of different soil AMF taxa at the genus and VT levels showed strong interactions between CC and water supply, which may partially explain the small effect sizes or lack of effects observed in the compositional analyses (Figure 4C and Figure 5). Such small changes, as well as the variability between field replicates, is consistent with the other AMF community measurements discussed above. Under no-CC, irrigated soils presented a higher proportion of the genus *Paraglomus* and lower of *Glomus* than drought soils (Figure 4C and Appendix A). Previous studies also reported the relative abundance of *Paraglomus* in soils or roots being favored by higher water availability [30,57]. Compositional analyses, however, found no clear abundance changes at the genus level (data not shown) and, thus, we cannot confirm that there was an absolute change in either *Paraglomus* or *Glomus*. The response of *Diversispora* VTs to water supply was among the strongest, and although they sometimes responded positively to both CCs and irrigation, strong interacting effects between these treatments and soil sampling depth were observed (Figure 5). Results in no-CC were similar to previous findings where *Diversispora* in roots were more dominant under lower precipitation conditions [58], although some *Diversispora* species were clearly favored by well-watered conditions in a microcosm experiment [30]. Unlike *Diversispora* VTs, we found some VTs belonging to the same genus sometimes behaved differently due to the interactions between CC, water supply, and soil depth (Figure 5). Likely, the interaction between these factors created unique environments that were more or less favorable to each species or VT.

The highest corn growth and yield were observed under irrigation and, within drought soils, under no-CC (Appendix A). Cover crops, especially those that survive the winter (e.g., rye, which was quite dominant in our CC), can limit mineral N availability for the cash crop [59]. We observed this only around the corn planting (April) for soil nitrate, with a similar trend for soil ammonium in April–June (Appendix A). This phenomenon could have been enhanced under drought conditions, where microbial activity and, thus, N mineralization, are restricted [59]. Notably, crop yield and stover biomass at maturity were negatively associated with AMF root colonization parameters, including the percentage of arbuscules. It is possible that the crop was more favored by a higher water or nutrient availability, while the higher AMF colonization observed in the drought treatments did not provide enough benefits, if any, to compensate for the poorer growth conditions. Even though AMF are known for their beneficial effects on crop growth and yield [12,60], this kind of result is not uncommon when studying native or resident AMF such as this one [23,61,62]. These studies are intrinsically more complex, with several interacting factors that may affect the outcome in terms of plant growth, including the AMF taxa colonizing the plant and their compatibility with the host in terms of functional performance (e.g., mutualistic vs. parasitic behavior) [23,63].

Overall, our results provide strong evidence that CCs can affect the structure of native soil AMF communities, even in the short term. Introducing CCs was also shown to modulate their response to changes in water supply levels, which are expected to be frequent in the face of climate change. This was mostly clear for soil AMF ASV composition and taxa, but also in terms of their capacity to colonize the subsequent corn crop. Even though the changes in AMF colonization were not directly reflected in the corn yields, the fact that both CCs and water supply were able to affect soil AMF communities in only one year of treatment suggests larger effects could be observed in the long term, potentially impacting the successful colonization and benefits of native AMF on cash crops. Our study provides valuable evidence of the response of AMF to sustainable agricultural practices in interaction with changes in water supply under field conditions and using next-generation sequencing, both of which are still quite limited in the AMF literature in spite of their relevance for agriculture. Our findings also revealed a strong influence of spatial heterogeneity (i.e., variability between field blocks) on soil AMF communities and, in some cases, their response to the applied treatments. This is something unexpected for field blocks, which are not supposed to interact with other independent variables. Thus, we recommend considering this aspect in the sampling design and metadata collection of field trials studying soil microbial communities. Future studies should also aim to relate shifts in soil AMF communities with AMF communities colonizing cash crop roots, in order to evaluate plant recruitment and establish more direct relationships between AMF composition and crop performance.

## 4. Methods

### 4.1. Site Description and Experimental Design

This study was carried out on a field trial established in the summer of 2017 on the VanArkel farm (42°33′57″ N 82°10′23″ W, 183 m a.s.l.), near Dresden, Ontario, Canada. Soils in the area, mainly from the series Tavistock and Maplewood, are characterized by being imperfectly to poorly drained, and fields had been tile-drained [64]. According to a preliminary analysis of 7 samples collected randomly before the trial, soils in the experimental site had a loam texture (30% sand, 47% silt, and 23% clay), pH = 5.9, 4.1% organic matter, and 46 ppm extractable phosphorus [64]. Before beginning the trial, the soils saw a diverse crop rotation with strip tillage and two CC cycles, as well as manure application. The crop rotation included sugar beet (*Beta vulgaris* subsp. *vulgaris*), soybean (*Glycine max* L.), winter wheat (*Triticum aestivum* L.), and corn (*Zea mays* L.). This study took place between fall 2017 and summer 2018, comprising the CC cycle and the corn growing season. Corn (Pioneer P0414) was planted on 1 May (74,568 plants ha^−1^, 76 cm row spacing) and harvested on 3–4 October 2018. This crop was side-dress fertilized with 168.2 kg N ha^−1^ in the form of urea ammonium nitrate (UAN) on 14 June. In all treatments, liquid hog manure (70,000 L ha^−1^) had been injected on 24 July 2017, two days before CC planting. The manure application was equivalent to incorporating 358.7 kg total N ha^−1^, 293.5 kg NH_4_-N ha^−1^, 203.6 kg P_2_O_5_ ha^−1^, and 365.4 kg K_2_O ha^−1^ [65]. All treatments were also subjected to strip tillage on 12 November 2017, to prepare the seedbed for corn.

The experimental design was a randomized split-plot with 4 blocks or field replicates. Two factors were tested: cover crops in the main plots (~3 by 30 m) and water supply in the sub-plots (0.77 by 6.1 m). Main plots were randomly arranged within field blocks and sub-plots within main plots. We compared two cover crop treatments: no cover crop (no-CC) and a 4-species mixture (CC) consisting of rye (*Secale cereale*), crimson clover (*Trifolium incarnatum*), Japanese millet (*Echinochloa esculenta*), and sunflower (*Helianthus annuus*). Following the winter wheat harvest (26 July 2017), CCs were no-till drilled at seeding rates of 28.0, 5.6, 0.9, and 2.2 kg ha^−1^, respectively. On 25 September, any remaining volunteer wheat in the no-CC plots was chemically terminated with glyphosate (Roundup^®^, Bayer, Leverkusen, Germany). One week before planting corn (24 April 2018), CCs were chemically terminated using 0.54 L ha^−1^ glyphosate.

The second factor evaluated in this study was the water supply during the corn growing season, which comprised two levels: drought and irrigated (i.e., ambient rainfall with supplemental irrigation). The sub-plots with the water supply treatments were placed along the central corn row of each main plot (4th out of 8 total rows), leaving gaps to avoid lateral water movement between treatments (~6 m downslope of the irrigation sites and ~3 m downslope of the drought sites). Drought conditions were simulated by covering the four inter-rows surrounding the central treatment row with vapor barrier plastic sheets installed on raised wooden troughs (installed on 19 June 2018) [65]. Using wooden pegs, a slight slope was created to allow water runoff from the sheets. These structures protected soils from both rainfall and stem flow, and the area covered by them ensured that the treatment row did not receive any lateral surface flow. The irrigation system consisted of a perforated aluminum pipe with a ~5 cm diameter and a ~6 m length attached to a 190 L rain barrel [65]. This pipe was installed at a 20 cm distance from the treatment row, and it was leveled to ensure a uniform flow rate. Irrigation levels were calculated based on the historical average weekly rainfall for Dresden, ON, Canada (~2 km from the experimental site) between 1981 and 2010, according to the Canadian Climate Normals database. According to this data, the average rainfall between 1 May and 30 September is ~20 mm week^−1^. Whenever the weekly rainfall levels were below this average, the difference was added to the barrels. The first irrigation event took place on July 6, because the June rainfall was above the 30 year average. The summer of 2018 was warmer than the 30 year average, and dry conditions were registered for several weeks around corn silking (mid-summer). The accumulated rainfall between early May and early September 2018 was 339.2 mm. More details concerning the experimental setup can be found in Ogilvie [65] and Ogilvie et al. [64].

### 4.2. Complementary Soil and Plant Data

To enrich the interpretation of our results, we used soil and crop data collected in these experimental plots. Soil mineral nitrogen was measured on samples collected at 0–30 cm depths. Briefly, ammonium and nitrate were extracted from the soil using 2 M KCl [66] and then measured in a SEAL Analytical AutoAnalyzer III (Mequon, WI, USA) [64]. The final mineral N concentrations (ppm) were calculated considering bulk density values [64].

Corn yield was measured from 10 plants located within 5 m long rows in each sub-plot. Harvested corn cobs were weighed before and after drying at 60 °C for several days to estimate both moisture content and dry weight [64,65]. The dried corn cobs were then shelled to calculate dry grain weight. The same plants were used to calculate aboveground corn biomass [64,65].

### 4.3. Soil and Root Sampling for AMF Analyses

Soil and root samples were collected on 24 September 2018, two weeks after the corn reached physiological maturity (7 September) and ~10 days before harvest (3–4 October). Both samples were collected from each sub-plot in the water supply treatment row. For each sub-plot, we collected five soil cores (⌀ = 2 cm), each comprising two depths (0–10 and 10–20 cm). The cores from each depth and sub-plot were pooled into a composite sample, transported in coolers and stored at 4 °C. The following day, samples were homogenized and a ~0.30 g aliquot from each sample was stored at −80 °C until the DNA extraction. By storing the amount needed for the DNA extraction separately, we avoided the drastic thaw event that would have occurred if weighing soils after freezing. Another soil aliquot was kept at 4 °C to calculate the gravimetric water content in each sample. In each sub-plot, root systems from two plants were dug up avoiding damage to the fine roots. Root systems from the different plants were kept separate, transported in coolers, and stored at 4 °C for 48 h. Following that period of time, roots were thoroughly washed with tap water, and fine roots were selected, tap dried, and stored at room temperature in 50% ethanol for microscopy analyses.

### 4.4. AMF Colonization of Corn Roots

To determine the AMF colonization of fine roots, we used a simple and low-hazard risk staining technique with ink and vinegar [67]. The first step consisted of clearing the roots with 10% (*w/v*) KOH for 25 min at 90 °C. After cooling for 30 min, roots were strained and rinsed several times using deionized water. Then, samples were soaked in vinegar for 60 min and boiled for 3 min in a 5% *v*/*v* ink–vinegar solution made with Sheaffer blue ink (A.T. Cross, Providence, RI, USA) and household vinegar (i.e., 5% acetic acid). Finally, each sample was rinsed using household vinegar and root fragments (10 cm length total) were aligned on a microscope slide parallel to its long axis. These root fragments were fixed to the slide using polyvinyl-lacto-glycerol (PVLG) and a coverslip. The stained roots were observed under 200× magnification, and AMF colonization was quantified using the magnified intersections method, based on the intersection between the roots and a hairline graticule in the eyepiece of the microscope [68]. The protocol by McGonigle et al. [68] was modified so that instead of totaling the arbuscules, vesicles, and hyphal structures present along the root fragment, we registered the presence or absence of any of these structures in each of the 100 fields of view observed. The percentage (%) of each different structure was calculated as the number of presences over 100 observations. Total colonization was based off of the percentage hyphal colonization. If hyphal structures were not observable but arbuscules and/or vesicles were present, then hyphae were also considered present, because these structures derive from hyphae penetration into the root cells.

### 4.5. Soil DNA Extraction and Illumina Sequencing

Soil DNA was extracted from ~0.30 g soil using DNeasy PowerSoil Kit (QIAGEN^®^, Hilden, Germany) according to the manufacturer’s instructions. DNA concentration and quality were measured using NanoDrop 8000 (Thermo Scientific™, Waltham, MA, USA). High-throughput sequencing of the AMF communities was carried out in an Illumina MiSeq platform (2 × PE300 bp) by Génome Québec (Montréal, QC, Canada). The optimization of the PCR conditions and template concentration, as well as library preparation and sequencing, were carried out by the sequencing facility. The forward and reverse primers chosen for AMF sequencing were WANDA (5′-CAGCCGCGGTAATTCCAGCT-3′) and AML2 (5′-GAACCCAAACACTTTGGTTTCC-3′), respectively [69,70]. These primers, which target a portion of the small subunit (SSU) or 18S rRNA gene (amplicon size ~452 bp), provide a higher taxonomic coverage and resolution of AMF than other primer sets [71].

### 4.6. Bioinformatics

Demultiplexed reads provided by the sequencing facility were imported to QIIME 2 2021.8 for bioinformatic analyses [72]. We used q2-dada2 [73] for denoising, dereplicating, trimming primers, filtering chimeras, and merging paired-end reads. This step generated a table with both nontarget and AMF amplicon sequence variants (ASVs). To retain only the AMF ASVs, we first carried out a taxonomic classification step using the latest MaarjAM reference database available (6 May 2019) [74]. This version of MaarjAM comprises a total of 457 VTs or phylogroups [74]. Before using this reference database, we carried out a series of cleaning steps to make it suitable for our dataset. Because several MaarjAM FASTA reads included the whole target sequence except our reverse primer (AML2), we first used *cutadapt* [75] to trim the reads at our forward primer (WANDA) only, excluding the reads that did not match the primer sequence and excluding those shorter than 300 bp. Then, we trimmed at the reverse primer to shorten the reads that were longer than our target but keeping those that did not match the primer. In this step, we also removed sequences that were too large (>600 bp). Using q2-RESCRIPt [76], the remaining reads were dereplicated in mode *uniq* (i.e., retaining identical sequences with differing taxonomies) and quality filtered (i.e., removing degenerate bases ≥ 7, homopolymer length ≥ 10). This clean reference database (31,034 reads in total) was used to assign a taxonomy to the ASVs using BLAST+ local alignment with q2-feature-classifier classify-consensus-blast [77]. The taxonomy was assigned at 97% sequence similarity and 95% query alignment coverage, following previous publications in AMF 18S rRNA sequencing [78,79,80]. After filtering out nontarget reads, one sample (from CC under irrigation at 0–10 cm) was removed from the analyses for presenting >90% non-AMF reads. The final table consisted of a total of 794,733 AMF reads and 249 AMF ASVs, which were used for downstream diversity and phylogenetic analyses. For the latter, a rooted phylogenetic tree was built using the pipeline q2-phylogeny align-to-tree-mafft-fasttree [81,82].

### 4.7. Data Analysis

Data were mostly analyzed using R v. 4.2.0 [83], except for the calculation of alpha diversity indices (i.e., within samples) and distance matrices for composition and beta diversity (i.e., between samples), which were carried out in QIIME2 2021.8 [72]. These diversity indices and distances were calculated on an ASV table rarefied at 9850 reads to normalize the number of reads per sample. To characterize alpha diversity, we calculated richness (including Faith’s phylogenetic diversity), evenness, and entropy indices both at the ASV and VT level. The response of individual variables (e.g., AMF colonization and alpha diversity indices) to CCs and water supply was analyzed using two-way ANOVA with linear mixed effect models in the R package “nlme” [84]. The split-plot hierarchy of the design (block > plot > sub-plot) was established in the random effects, while the variables of interest and their interaction were considered fixed effects.

Changes in soil AMF composition were evaluated using quantitative and qualitative phylogenetic distance metrics (weighted and unweighted UniFrac, respectively) [85]. The response of AMF composition to CCs and water supply was tested using permutational multivariate ANOVA (PERMANOVA) with *adonis2* in the R package “vegan” [86]. Before this analysis, we analyzed the variability within each treatment (beta dispersion or beta diversity) with *betadisper* in “vegan”. Distances were visualized using nonmetric multidimensional scaling (NMDS) with the function *metaMDS* in “vegan”.

The differential abundance testing of AMF taxa (VT, genus, and class levels) was conducted with a compositional approach using ANOVA-like differential expression (ALDEx) in the package “ALDEx2” [87]. Those virtual taxa with very low abundance (i.e., less than 20 reads in total) and/or prevalence (i.e., present in less than 2 samples) were excluded from the analysis. As recommended for small sample sizes, we used the ALDEx effect size instead of the corrected *p*-values to determine if a specific taxon was sensitive to the tested factor [88]. Because we expected treatment interactions, in addition to carrying out a global analysis, we evaluated CC effects within each water supply level, as well as water supply effects within each CC treatment. Changes in the clr-transformed relative abundance of VTs were visualized on a heatmap using “gplots” [89]. Centered log ratio-transformed abundances were also used in any downstream analyses involving taxa. Unique and shared taxa were checked and visualized using InteractiVenn [90]. 

## Figures and Tables

**Figure 1 plants-12-01015-f001:**
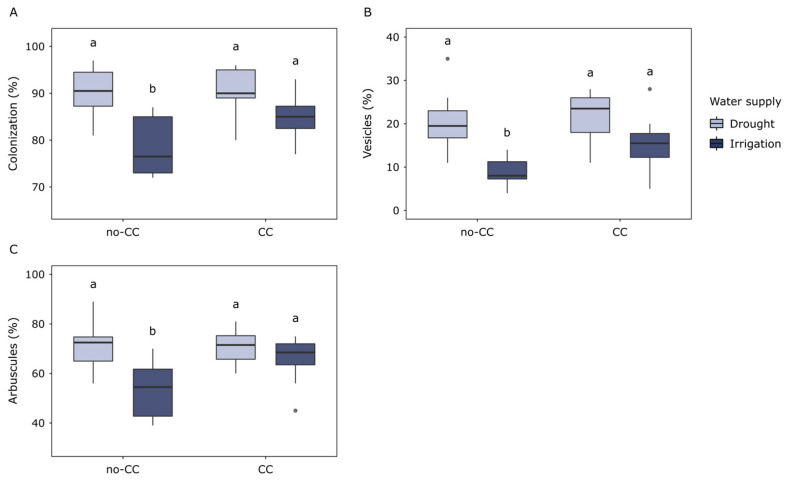
Arbuscular mycorrhizal fungi (AMF) colonization of corn roots in response to cover crops (no-CC: no cover crop; CC: 4-species cover crop) and water supply (drought; irrigation). Data for hyphae colonization (**A**), vesicles (**B**), and arbuscules (**C**) are shown. Different letters show significant differences between water supply treatments for each CC treatment (Tukey test, alpha = 0.05). Boxplots show median (middle line), 25th and 75th quartiles (hinges), minimum and maximum (whiskers), and outlying points individually. Full ANOVA results are available in Appendix A.

**Figure 2 plants-12-01015-f002:**
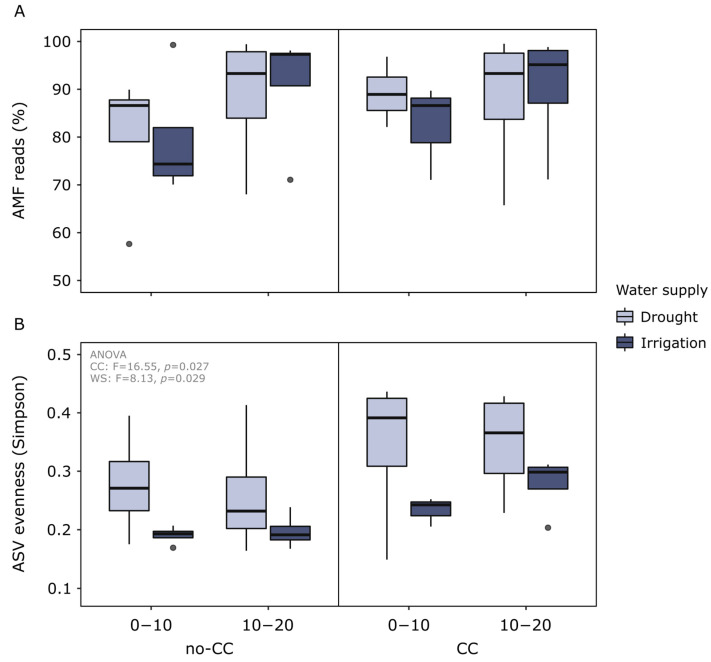
Percentage of arbuscular mycorrhizal fungi (AMF) reads out of the total sequenced reads (**A**) and amplicon sequence variant (ASV) Simpson evenness of soil AMF communities (**B**) in response to cover crops (CC) (no-CC: no cover crop; CC: 4-species cover crop) and water supply (WS) (drought; irrigation). Data are shown for two different soil depths (0–10 and 10–20 cm). Boxplots show median (middle line), 25th and 75th quartiles (hinges), minimum and maximum (whiskers), and outlying points individually. AMF reads were not significantly different between the treatments, whereas AMF ASV evenness was higher in CC than no-CC and higher in drought than irrigation (ANOVA F- and *p*-values are shown in the plot).

**Figure 3 plants-12-01015-f003:**
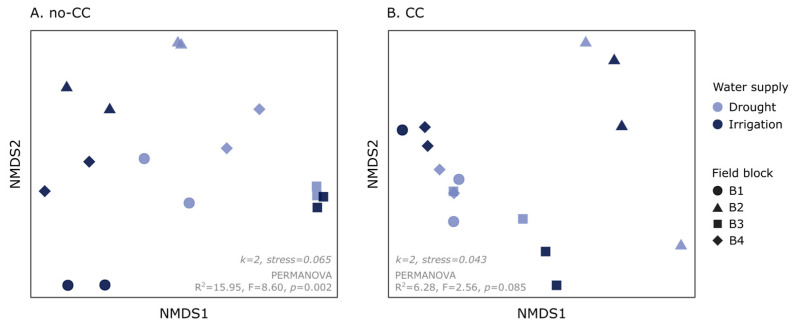
Nonmetric multidimensional scaling (NMDS) showing changes in soil arbuscular mycorrhizal fungi (AMF) community composition (weighted UniFrac) in response to water supply (drought, irrigation). The results are shown separately for each cover crop treatment: no cover crop (no-CC) (**A**) and 4-species cover crop (CC) (**B**). Each plot also shows the NMDS dimensions (k) and stress values, as well as the water supply effects according to PERMANOVA (R^2^ = % variance explained, F-value, *p*-value). The full PERMANOVA results can be found in Appendix A.

**Figure 4 plants-12-01015-f004:**
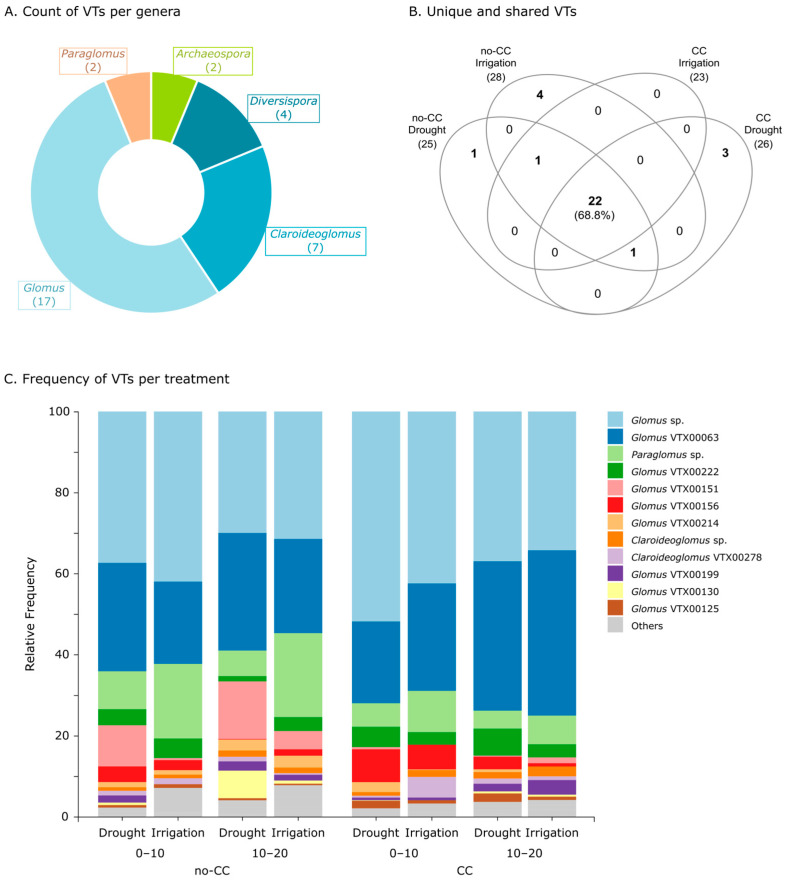
Pie chart showing the number of arbuscular mycorrhizal fungi (AMF) virtual taxa (VTs) per genus (**A**); Venn diagram showing unique and shared AMF VTs between treatments (**B**); and taxa bar plots showing the relative frequency of AMF VTs in each treatment (**C**). In (**B**,**C**), data are shown for both cover crop (no-CC: no cover crop; CC: 4-species cover crop) and water supply (drought; irrigation) treatments. In (**C**), data is also shown by soil depth (0–10 and 10–20 cm), and only the highest abundance VTs are shown, while the rest are collapsed into the “Others” category. Taxa bar plots at the genus level, both pooled and by individual field replicate, can be found in Appendix A.

**Figure 5 plants-12-01015-f005:**
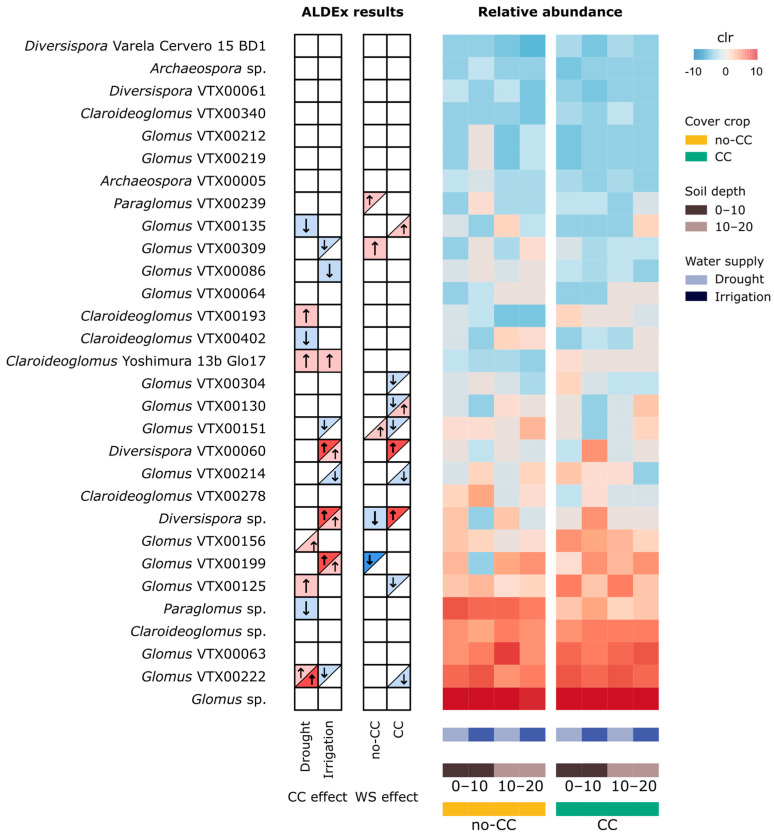
Changes in the relative abundance of soil of arbuscular mycorrhizal fungi (AMF) VTs in response to cover crop (CC) and water supply (WS) treatments at two different soil depths (0–10 and 10–20 cm). The table on the left summarizes the differential abundance testing results (ALDEx2) for CC and WS at each level of WS and CC, respectively. In the left side table, arrows and colors indicate an increase (red, ↑) or decrease (blue, ↓) in response to WS (i.e., irrigation) and CC. When effects were only observed at one soil depth, only half of the box is shaded (upper left: 0–10 cm; lower right: 10–20 cm). Taxa with |effect size| > 1 are shown in darker colors and thicker arrows on the table; the rest of the highlighted taxa have |effect size| > 0.5 but <1. In the heatmap, VT relative abundance is expressed as centered log ratio (clr).

## Data Availability

The raw sequences were deposited as BioProject ID PRJNA919021 in the Sequence Read Archive (SRA) of the National Centre for Biotechnology Information (NCBI).

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
