# Peer review of "Cover Crops Modulate the Response of Arbuscular Mycorrhizal Fungi to Water Supply: A Field Study in Corn"

_plants, 2023, doi:10.3390/plants12051015_

Round 1

Reviewer 1 Report

The manuscript with the title ''Cover crops modulate the response of arbuscular mycorrhizal fungi to water supply: a field study in corn'',  is an interesting study that well-covered the interaction of AMFs, cover crop systems soil functionality, and water availability in a single work. The current study is a good example of well-designed research that is precisely written which definitely indicates the author's dominance on the subject.

The introduction provides the background of the research well. Material and methods are well-described specially AMF processing and analyzing. The discussion section also properly covered the subject and the authors well interpreted the findings.

Overall, there isn't much more that can be said; except suggesting it for publication!

Author Response

Dear reviewer,

We want to thank you for the time you dedicated to reviewing our manuscript and we are pleased that you found it satisfactory and ready for publication.

Kind regards

Reviewer 2 Report

Dear Authors,

I think your manuscript needs improvements before publication. Altogether, I confirm that the work is novel and the approach is interesting, yet, the following points require your attention. Please address.

Major issues.

1., Block effect. Typical to field studies, there are major differences among the actual points (sites, “blocks”). I think the nice stress values and the alignment of points in Fig.3. suggest that NMDS actually separates the sites rather than your treatments.

1a., Though if a phenomenon is to be supported by statistical evidence, a pooled dataset is most appropriate, perhaps you should consider splitting your dataset along the block factor and submit the numbers for each block to statistical tests.

1b., Plots showing blocks separately might give additional insight into what is going on; I think this is especially relevant for Fig.3., Fig. 4C, and Fig. S3.

2., p-value adjustments. You did many statistical tests, but no p-value adjustment was carried out to obtain an overall false discovery rate of p = 0.05 (Broadhurst & Kell, 2006). I suggest using the Benjamini-Hochberg procedure, which provides decent sensitivity. This will likely resolve the issue outlined in P162-163 (in other words, I think P = 0.028 might have been a statistical significance by chance only). The same p adjustment issue is important regarding changes shown in Fig. 5 – are “changes” statistically significant?

Minor issues.

L212-213: after clr transformation? (Gloor et al., 2017)

L263-265: This is not surprising with an AMF specific primer in my opinion.

L272: Did references 17,20-22,48 also terminate the cover crop with glyphosate? Can that be behind the difference?

L277: lack of leguminous?? What about Trifolium?

L399-400: does this texture apply to all the blocks? Please clarify!

L417-418: Add authority names for botanical taxa on first mention.

Fig S4: a composite barplot would be better (it is hard to decipher the actual numbers)

Some abbreviations should be defined on first use. This also applies to ones widely used in the field, such as ASV and VT.

Add sequencing parameters for HTS.

Best regards.

References

Broadhurst, D. I., & Kell, D. B. (2006). Statistical strategies for avoiding false discoveries in metabolomics and related experiments. Metabolomics, 2(4), 171–196. https://doi.org/10.1007/s11306-006-0037-z

Gloor, G. B., Macklaim, J. M., Pawlowsky-Glahn, V., & Egozcue, J. J. (2017). Microbiome Datasets Are Compositional: And This Is Not Optional. Frontiers in Microbiology, 8. https://www.frontiersin.org/article/10.3389/fmicb.2017.02224
